# Secured MCDM Model for Crowdsource Business Intelligence

**Musiri Kailasanathan Nallakaruppan [1,\*]**, **Ishita Johri [1]**, **SivaRamaKrishnan Somayaji [1]**, **Surbhi Bhatia [2,\*]**, **Areej A. Malibari [3]** and **Aliaa M. Alabdali [4]**

1   School of Information Technology and Engineering, Vellore Institute of Technology, Vellore P.O. Box 632014, India
2   Department of Information Systems, College of Computer Science and Information Technology, King Faisal University, P.O. Box 400, Al-Hasa 31982, Saudi Arabia
3   Department of Industrial and Systems Engineering, College of Engineering, Princess Nourah Bint Abdulrahman University, P.O. Box 84428, Riyadh 11671, Saudi Arabia
4   Faculty of Computing & Information Technology, King Abdulaziz University, P.O. Box 344, Rabigh 21911, Saudi Arabia
*   Correspondence: nallakaruppan.mk@vit.ac.in (M.K.N.); sbhatia@kfu.edu.sa (S.B.)

**Abstract:** In the current era, there are a plethora of mobile phone companies rendering different features. It is challenging to distinguish the best and create correlations among them. However, this can be accomplished through crowdsourcing. Crowdsourcing is the process of gathering information from multiple sources, and we use the AHP (Analytic Hierarchy Process) process to determine which company's model is the best among many. The weight value of each model is compared to the assigned values, and if one of the company product weights is greater than the assigned weight, that product is the best. Eventually, we can use this process to select the most preferred and best mobile phone model from among all other models. Gray Relational Analysis (GRA) is one of the most popular models, employing a grey co-efficient that estimates the data items by ranking. This model defines a process's situation or state as black with no information and white with perfect information. In this work, AHP initially assumes criteria weights and assigns rank with the CR (Consistency Ratio) of 1.5%. The criteria weights are re-assigned based on the outcomes, and the CR remains constant as 1.5%. This work also provides an environmental-based attribute access control system, which adds the strength to the system by providing security and the integrity. So, this proposed work performs as a decision support system combined with the security enhancements, and hence it becomes a complete framework to provide a solution to a target application. The novelty of the proposed work is the combination of the crowdsourcing with the recommender system on a secured framework.

**Keywords:** crowdsourcing; AHP; GRA; access control; recommender system; decision-support system

## 1. Introduction

Recommendation systems help users find meaningful and actionable information from large amounts of unorganized data. To meet a user's needs or quickly find information that might be of interest to the user, these systems [1] filter and analyze vast amounts of information to help a particular user purchase a product. Users' preferences can be used to predict and identify certain items. They are successfully used to support various decision-making processes, which helps people to find content of interest among the many options available. Crowdsourcing is the process by which the users can obtain goods, services and ideas from a large group of people. It divides the work between the workers to simplify the process and achieve the result easily. A critical need for real-time recommendations has emerged in crowdsourcing systems. On the one hand, users want effective recommendations of the top-most phones in terms of their use and features, and on the other hand, requesters want trustworthy recommendations of the best mobile





phone models for their tasks in terms of cost and warranty. There is a difference between crowdsourcing and outsourcing. Crowdsourcing comes from less specific and more public groups, whereas outsourcing comes from specific or named groups. The Internet is the main source of crowdsourcing because users can obtain more sources and ideas from the internet. The advantages of crowdsourcing include speed, quality, scalability and flexibility. By crowdsourcing, the users obtain the weight value of the products. The weight value of the company's product is checked with the assigned weight value. The weights are then compared; higher assigned weights provide the best product, while those with a lower assigned weight value are not the best products. As part of the decision-making process, the AHP helps by calculating the weight values. The weight values of the products collected via crowdsourcing which are then inputted to the AHP process to determine which is the best product among the different company products [2]. Such a structure is susceptible to deliberate misuse of the system's resources, including asset destruction, theft, modification, invasion of privacy, interruption of operations, unauthorized use of assets and even bodily harm to personnel who have dataset rights. There is always danger, even if the company and all of its systems abide by the contracts, rules and regulations that may be relevant. Hence, a system such as this needs to be protected against any interference [3]. We propose a far-reaching access control technique that permits record owners to peruse and compose information while likewise upholding system-level encryption-related exercises. It is an extensive system which permits jobs to be coordinated progressively, in addition to fostering the job based admittance control model. Less advantaged positions are found at the lower part of the ordered progression, while the most elevated jobs are found at the top.

### 1.1. Contributions of the Paper

The system's goal is to close the gap between recommender systems and crowdsourcing. Depending on the recommendation algorithm and context settings utilized, it will employ various recommendation methods, apply them in accordance with the present situation of known weights, and provide various alternatives for privacy-preserving recommendations. The proposed methodology maps explicit EBAACM model jobs to explicit arrangements of qualities, and higher jobs can acquire all authorizations from lower jobs. Here, jobs basically refers to the role or position a user holds. Various external attributes are also taken into consideration. This creates a safe transaction system for each digital identity of the user. Therefore, the system has comprehensive security for its data. The crowd sourcing and recommender systems are combined in this research, resulting in the proposed advance system and its success. This work also includes the provision of an environmental-based attribute access control system, which contributes to the robustness of the system by guaranteeing both its safety and its integrity. Therefore, the work that is being presented functions as a decision-support system, and when combined with the security advancements, it turns into a comprehensive framework that could serve as a solution to a specific application. The innovative aspect of the work that has been proposed here is the coupling of crowdsourcing with a recommender system within a protected environment.

### 1.2. Organization of the Paper

The format of the paper plan is as follows: An introduction to crowdsourcing and mobile sales prediction model can be found in Section 1. Section 2 provides an overview of the many prediction techniques employed by recommender systems in a variety of situations, some of which may involve crowdsourcing. With AI being used in the articles, Section 3 focuses on recent research on crowdsourcing. Analytical hierarchy procedure and gray relation analysis is covered in Section 4; the experimental results are covered in Section 5. According to observations and suggestions, Section 6 introduces the model's safety and trustworthiness and also concentrates on the algorithm and how it contributes to strong security. Section 7 concludes the paper with a quick overview of its main ideas, suggestions for future investigations and other related observations.

### 1.3. Roles of AHP and GRA

Due of the multi-dimensionality and units of choice criteria, this research used fuzzy and rough number MCDM methods. It may be difficult to quantify the many aspects of a phone's design that play a role in deciding which model is ideal, since these aspects often exist in more than one unit or dimension. Distinct membership functions in rough and fuzzy numbers eliminate bias in decision making. During the judging process, fuzzy and rough numbers decrease bias but increase computational labor. Fuzzy AHP (F-AHP) applications have been a milestone in evaluating weights for choice criteria as a result of the computational approach of a Fuzzy System. Fuzzy GRA (F-GRA) has also been effective because of its capacity to create comparable sequences and provide the sense of comparing options with a reference series [1]. The AHP provides subjective analysis where the opinions are acquired from experts in the relevant field of interest. On the other hand, the GRA is a completely independent objective-based ranking solution. This does not depend upon the weight importance, but only depends on the min–max composition and gray relation between them with the other values in the dataset. In the proposed work, we apply both AHP and GRA to ensure that the balance is achieved in the prediction of the subjective and objective analysis of these fuzzy systems. The final criteria importance of GRA is estimated through sensitivity analysis and the corresponding importance is re-assigned in AHP as weight importance. The consistency ration of AHP is compared before and after the change in criteria weights. If it remains the same, then the results are validated for the problem.

### 1.4. Advantages of AHP and GRA

There are various advantages in both AHP and GRA. These advantages are listed below in the different subsections.

#### 1.4.1. Advantages of AHP
- Simplicity.
- Versatility.
- Many criteria can be selected.
- Easy to use.
- Consistency check for working on collaborative projects.

#### 1.4.2. Advantages of GRA
- Weight independent and AHP independent.
- Complete objective solution.
- Simple and easy to apply.
- No limitations on the sample size
- Normal distribution of data.
- Simple computation.

## 2. Literature Survey

In recent years, it has become important to determine the index weight reasonably. Otherwise it will have a bad influence on the objectivity and authenticity of the results. AHP and GRA processes are used to find the weight and then the best one is selected [4].

Based on the earlier data gathered by the platform during crowdsourcing, the researchers derive an incentive management approach based on mathematical optimization that executes business processes in a cost-optimal way considering their deadlines [5]. Crowdsourcing is a new business concept which involves different integrations of a job provider, a crowd worker and a crowdsourcing web-based platform, and it describes the existing business model of crowdsourcing and critical analysis for each business model [6]. The lightweight bricks are in high demand in the market. To meet this demand, they use AHP process and a decision tree for production decision making. AHP has a four-step developing hierarchy such as normalization, creating the pairwise comparison matrix, weight

estimation and the synthesis and logical consistency test [7]. Enterprise brand concerns are between both the producer and the customer in the market, since the branding process issues are the perceived quality, brand knowledge, brand credibility, brand image and brand identity [8]. Service industries meet many intangible factors with tangible objects. Intangible factors such as innovative ideas, new service attributes, learning principles and self-service technologies have a great impact on business success and customer satisfaction, so the AHP and GRA is used to assign the local and global hierarchical priorities among the different categories of service quality attributes [9,10]. In this study, the authors use crowdsourcing to gather the information, and then the AHP process and GRA process is used to divide the large quantity of gathered information into small sub elements and identify which is the most preferable company using the weights.

In [11], presents a label aggregation strategy for hierarchical classification techniques. To demonstrate the value of introducing a hierarchical organizational structure and to increase the accuracy of label aggregation, an experiment was conducted using a real crowdsourcing problem of hierarchical classification. To create a more precise and nuanced interpretation, the proposed method in [12] shows how crowdsourcing, service experience visualization, and cluster analysis can all be used together. A cluster analysis of different values was used to find groups of participants who differ in their confusion and payment method ratings. Three of the five clusters found contained individuals exhibiting susceptibility to different types of disorders. This was surprising, as internal research from the telecom companies showed that the level of annoyance was greatly influenced by whether or not customers paid for the service. However, one of these three clusters contained individuals who were not sensitive to payment differentials and were relatively reconciled with non-collection. The interactive context-aware recommender system established by this study [13] advances the concept of human–computer interaction in traditional CAR. A car rental website built using the proposed iCAR technology is shown as a demonstration to confirm its feasibility and usefulness. The same information is used in this iCAR to help users find the vehicle that best suits their needs. iCAR used three-dimensional information, such as user, item and contextual information to enhance the accuracy of traditional CAR systems and provide users with more accurate recommendation results. The authors also plan to apply it to other industries, such as online shopping and travel package recommendations, to improve the results.

In [14], the authors describes the development, application and evaluation of ForeXG-Boost, a car sales forecasting tool with high forecasting accuracy which requires little computing power. After exhaustive research to assess the impact of various variables on vehicle sales through information gathering and data association, the most meaningful items from the feature set are selected for prediction. Numerous tests confirm ForeXG-Boost's ability to combine low overhead with high prediction accuracy. In [15], the authors describe the Sturgis formulation, which is used to replace the iterative behavior of fuzzy logic within the prediction method. AFER (average forecasting mistakes rate) and SSM (suggest squared mistakes) are the metrics used. The effects showed that the ANFIS set of rules outperformed the bushy time series with AFER values much less than 15% and errors greater than 20%. ANFIS-MSE values notably decreased compared to fuzzy time series.

The research carried out by [16] aimed to reveal patterns of influence of certain factors on car sales in the European market and functional patterns for predicting future sales. Global aggregate indicators and indicators for specific regions or clusters were used to build mathematical prediction models and linear regressions. This means that applying the model to the test sample data yields good predictive accuracy. In [1], the authors propose the Gaussian process latent variable model factorization (GPLVMF) method, which is a nonlinear dimensionality reduction technique. The GPLVMF model is solved using a variational inference approach. There are two contributions when it comes to setting up the recommender system: fixing the real-valued latent space to use a real-valued context and handling the bias effects by setting a non-zero mean function.

The choice of a relevant machine tool is the most crucial challenge in the manufacturing industry, as it will affect the overall performance. The present multi-criteria decision-making (MCDM) method for choosing machine tools mostly emphasizes the subjective viewpoint. Wang et al. [17] deployed a modified hybrid MCMD model for the right machine tool selection. First, the proposed method uses an integrated weight strategy that combines subjective weights obtained through the fuzzy decision-making trial and evaluation laboratory (FDEMATEL), with objective weights gained through entropy weighing (EW). Then, the authors apply defuzzification VIKOR to obtain the ranking choice.

Due to the overwhelming number of service providers (SP) in crowdsourcing, the management of information is a rising issue. Shixin Xie et al. [18] intend to present an evaluation framework for quality of service (QoS) for SPs in KI-C to accurately and thoroughly characterize the QoS of SPs, which can help with the effective selection of qualified SPs. In order to use the collective intelligence of a large number of people, a task may be outsourced to an online marketplace for crowdsourcing. The crowdsourcing process saves time and money because of concurrent task execution and online labor markets. Everyone finds it difficult to choose the right jobs with the relevant labels and give them to the right workers during a crowdsourcing exercise. In this study [19], the authors have suggested a mechanism for allocating the task to the workers. To give the task to the most qualified worker, a multicriteria-based task assignment (MBTA) mechanism is developed. This mechanism employs methods for allocating weight to the factors and rating the workers. 5G technology, which is widely employed in the domains of medicine, transportation, energy and other industries, has developed quickly in recent years. 5G base stations, which are the essential components of the 5G network, enable wireless signal transfer between wired and wireless terminals, as well as wireless coverage. However, issues such as poor user experience and limited coverage area commonly arise as the number of 5G base stations gradually grows. Therefore, it is essential to assess the overall performance of 5G base stations in order to identify any issues that may have arisen during base-station installation. First, the operational performance, financial performance, environmental effect and social influence perspectives are used to build the performance evaluation [20] index system. Then, a unique hybrid multicriteria decision-making (MCDM) model based on the difference-quotient gray relational analysis (DQ-GRA) technique and the Bayesian best–worst method is implemented.

*Literature Summary*

A summary of relevant existing research works is provided in Table 1. In any Information and Communication system, authentication and authorization [21] are the major metrics to be taken care of to achieve an accurate performance of the system. Thus, in our proposed work, we have deployed an attribute-based access control system for better working efficacy. Recommender systems are typically information-filtering systems used to suggest relevant information to a user. There are various context-aware recommendation systems which are used for decision making in different scenarios [13]. This proposal uses a recommendation system to select the best mobile model based on various decision-support methods.

**Table 1.** Summary of existing works.

| Ref. No | Methods | Advantages | Research Challenges |
|---|---|---|---|
| [5] | Framework for sustainability of crowdsourcing in business process | A well-interconnected framework for crowdsourcing from the business perspective | Lack of an identity-management policy for access control |
| [13] | Context-aware recommender System | Providing the appropriate car for the users using contextual information | The system lacks sensitivity analysis |
| [22] | A smart motion detection approach based on gray relational analysis is applied | Efficient motion detection performance is achieved | Overhead cost for video rendering has to be considered |
| [23] | Best-only method is deployed to overcome the challenges in choosing the most relevant cloud service provider | The method is better than AHP and BWM for reducing computational complexity | The method does not provide a recommendation based on resource cost and availability zones. |
| [21] | A modified Role-based Access Control model is deployed, along with deep reinforcement learning | The access control policy is dynamic so that the users adhere to the security standards | The method fails to address the computational overhead for the access policy deployed. |
| [20] | A hybrid multicriteria decision-making (MCDM) model based on the Bayesian best–worst method (BBWM) is implemented for the performance of 5G base stations | Better coverage range with less cost of infrastructure | Signal interference and authorization is not considered |

## 3. Recent Research on Crowdsourcing with AI

With the use of the internet, a large number of individuals can contribute information to a task or project through the practise of crowdsourcing.

Depending on the project, crowdsourced contributors can be either compensated or gratuitous. However, in the realm of artificial intelligence, most crowdsourcing work is an automated service. In machine learning, crowdsourcing comprises allowing the mass deployment of products and services. For machines to execute NLP and NLU tasks such as classification, feature engineering and decision-support system and text categorization, data labeling is helpful. The comparative analysis of various applications is presented in Table 2.

**Table 2.** Comparative Analysis of the proposed work with Recent developments of the Crowd-Sourcing Applications.

| S. No | Application | Technology | Strength | AI Usage | Security | Ranking |
|---|---|---|---|---|---|---|
| 1. | Crowdsourcing with deep learning | Crowdsourcing for Learning of annotators | learning through annotator with expertise inferring true labels from meager annotations. | Deep Convolution neural networks | No | No |
| 2. | Crowdsourcing for gesture control in smart phones. | Crowdsourcing for analysing the tapping features of the mobile phones | Determining the tapping feature of a mobile by aggregating data using surveys and crowdsourcing. | Gesture control and sensing | No | No |

**Table 2.** *Cont.*

| S. No | Application | Technology | Strength | AI Usage | Security | Ranking |
|---|---|---|---|---|---|---|
| 3. | Crowdsourcing for crop yield | Crowdsourcing with Image sensing through CNN | Sentinel-2 with remote sensing imagery | CNN with sentinel-2 for remote sensing of images | No | No |
| 4. | Proposed system | GRA for grading, AHP for weight training, attribute-based access control security mechanism in Crowdsourcing | Recommendations with AHP, ranking with GRA, security with attribute-based access control mechanisms | AHP and GRA for ranking and recommendations | Yes | Yes |

### 3.1. Crowdsourcing with Deep Learning

With the advancement in the web, most websites and social media use crowdsourcing to curtail the time and cost. The authors of [24] implemented a Bayesian network embedded with deep learning (DL) when inculcating the learning from the target crowd from different annotators. This framework uses low-rank structure in annotations. For learning, each annotator's expertise is used to infer true labels from meager annotations. After inferring the true labels, the DL model is trained to improvise the learning process.

### 3.2. Crowdsourcing for Gesture Control in Smart Phones

Nowadays, smartphones have features which render better user interaction. Some smartphones use gesture-based inputs; tapping is one of the most frequently used features. Mobile vendors are striving hard to develop these gesture controls in accordance with user preferences. The authors of [25] developed a method for determining the tapping feature of a mobile by aggregating data using surveys and crowdsourcing. The authors implemented a deep neural network upon the data collected to find out users' preference between a tappable and a non-tappable element. The authors also deployed a tappable technique called TapShoe, which auto-identifies disparities of tappability feature of an element.

### 3.3. Crowdsourcing for Crop Yield

Remote sensing imagery data are used to detect the plant health in a crop field. The challenge of this imagery data for creating crop-type maps is the deficiency of ground-truth labels. The work in [26] deployed convolution neural networks (CNNs) to generate the crop-type map using crowdsourced data, Sentinel-2 and remote sensing imagery. The farmer-generated images were used to calculate the crop-type labels which were inputted to the CNN model. However, the data had lot of noise and lacked location accuracy. After this, the data were pre-processed and the imagery data from remote sensing units were deployed in CNN for training to generate points in the crop field. The authors concluded that the CNN differentiates rice, cotton and "other" crops with 74% accuracy and provides better results than the random forest method.

### 3.4. Comparative Analysis

Three recent related research works were compared with the prescribed work. The first work is on the Bayesian network embedded with the Deep Learning for annotation purposes. The second work under comparison was gesture detection using artificial intelligence, and the third work under comparison was image analysis through sentinel-2 imagery. Apart from the accuracy levels, the prescribed work provides recommendations for privacy and security through an environment-based attribute access control model which provides recommendations for privacy and security in a crowd-sourcing domain,



which is the prime highlight of the work. Thus, the prescribed work is a multi-dimensional solution in comparison with recent works on crowdsourcing.

## 4. Architecture of the System

The smartphone market is expanding worldwide due to a variety of factors, including rising disposable income, the development of telecom infrastructure, the appearance of smartphones designed with affordability in mind and an increase in the number of product launches. It is challenging for the user or buyer to stay true to his demands and purchase a suitable phone while taking into account all the characteristics required, what with so many models with such a variety of features entering the mainstream. A strong framework that essentially operates by giving weights to the features that a customer might take into account while making a decision is recommended as a solution to this problem. Using the AHP (Analytic Hierarchy Process), we were able to identify the best phone model after compiling all the pertinent data from various online resources. The system's decision is based on a fuzzy weights system in which the weight value of each company's product is compared to the weight value that has been assigned to it. The comparison identifies the top model at an intermediate level. One of the most popular models of gray system theory is the Gray Relational Analysis (GRA), which Deng Julong created. In this instance, GRA makes use of a particular idea: dividing the data into two categories, black or white, and identifying any areas of uncertainty that could be resolved effectively. Five attributes—name, model, pricing, feature and warranty—are taken into account throughout the recommender system's decision-making process when employing calculated weights. The more significant quantitative characteristic that makes up the standardized matrix is determined by pairwise comparison and AHP scaling. In addition, GRA aids in the creation of the normalization and deviation matrices and the calculation of the gray relationship coefficient. For the intelligent decision support system, priority is determined in terms of a percentage and is ranked in order of priority. Sensitive data are processed by the system, making them susceptible to different cyberattacks and modification risks. Consequently, a role-based framework is advocated for system security. Users can range in privilege from low roles to high roles. Additionally, this confirms whether the environment in which the system functions has static or dynamic features. To provide a safe environment, the robust framework first employs the idea of a digital identity number and cryptographic operations. A unique identification is then generated for each operation and each stated role in the hierarchy using hashing techniques, maybe followed by double hashing. Key certificates produced by Certified Authorities (CA) and Proxy, which essentially serves as a semi-trusted server for data encryption and re-encryption, are additional functionality aspects of the system. It has a PKI key and an X.509 certificate to authenticate other system modules, which elaborates operation. Furthermore, a decision is made based on the role environment, which may include details about the external conditions, such as access times, working and non-working hours, geographies and other dynamic elements of the access-control scenario that may be paired with system properties. The overall architecture of the system is presented in Figure 1.

### 4.1. Analytical Hierarchy Process

The Analytic Hierarchy Process (AHP) is based on dividing a highly advanced problem into small sub-elements, then organizing and combining them to form an orderly hierarchical structure, and determining the relative importance of the elements through pairwise comparison and standardized comparison [27].

$$y = \sum [a_j w_j] - n/wl \tag{1}$$

AHP is a decision-making process which is used to decide the best product with the help of the previously outlined criteria [28]. Here, we will select the best product based upon the weights of the criteria which have been calculated. We have taken five criteria in this paper, i.e., 1. Name, 2. Model, 3. Features, 4. Cost, 5. Warranty.

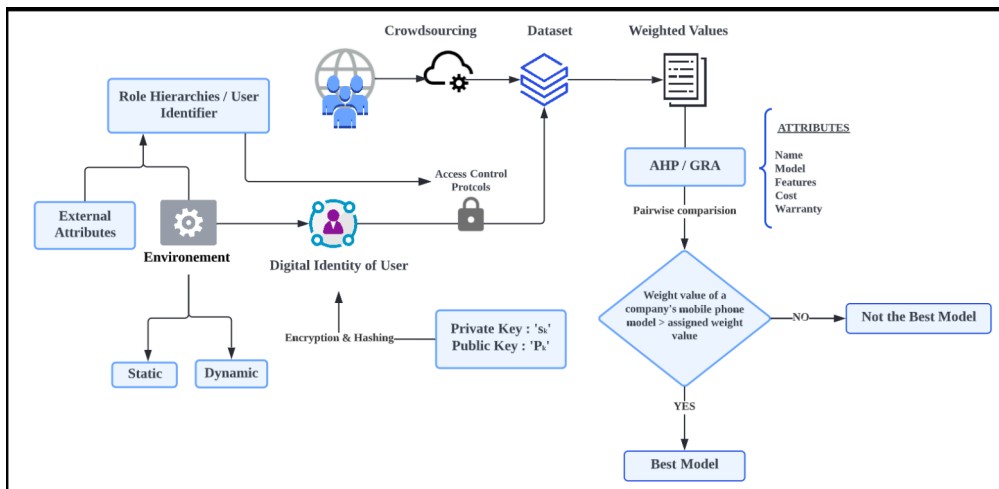

**Figure 1.** Overall Architecture of the System.

### 4.2. Gray Relational Analysis

Gray Relational Analysis (GRA) is also called Deng's Gray Analysis Model [22,29]. It is one of the most widely used models. It uses a specific concept of information. This model defines the situation or state of a process with no information as black and with the perfect amount of information as white. The gray system has part of the information known and part of the information unknown. The good and quality information is the process in which black changes to gray and then to white. However, uncertainty [30] always exists, in the middle, somewhere in the end, and somewhere in the gray area [31]. In some processes, without information, no solution can be defined for a system. Meanwhile, a system with information has a unique solution. Gray analysis does not provide the best solution, but it does provide the techniques to determine the good solution. The GRA model is a good solution to real-world problems [32].

The Gray Relational Analysis has three steps: 1. Normalization matrix, 2. Deviation sequence matrix, 3. Gray Relation Coefficient.

In this paper, we have collected the dataset, which consists of one hundred attributes. We have taken the same criteria as in the AHP, i.e., Name, Model, Features, Cost and Warranty. Then, we have calculated and analyzed the maximum and minimum values for the criteria in the collected dataset. After that, we have calculated the normalization matrix. By using the normalization matrix, we calculate the deviation sequence matrix; lastly, we have calculated the gray relation coefficient.

In the dataset which we have collected, there are many mobile products with different types of models, features, cost and warranty. It is hard to choose which is the best product among them. However, by using the GRA process, we can normalize the dataset and we can find the best product [23]. The best product is selected based on the rank. The mobile product which is ranked first among all of the products will be considered the best mobile product.

## 5. Results

The result section contains Expert Review, Pair-wise comparison, Normalization, Weight estimation and ranking for AHP. This section also estimates the normalization, estimation of the grey relation co-efficient, estimation of gray relation grades and ranks for GRA. The simulation of the selection of alternatives is presented in the Figure 2. The Expert review based importance and the corresponding scale is presented in the Table 3.

### 5.1. Steps of AHP Experimentation
#### 5.1.1. Weight Normalization of AHP

The weighted normalized matrix in Equation (1) and the individual attributes $a_j$ are trained with weights $w_j$ and normalized based on the order of importance.

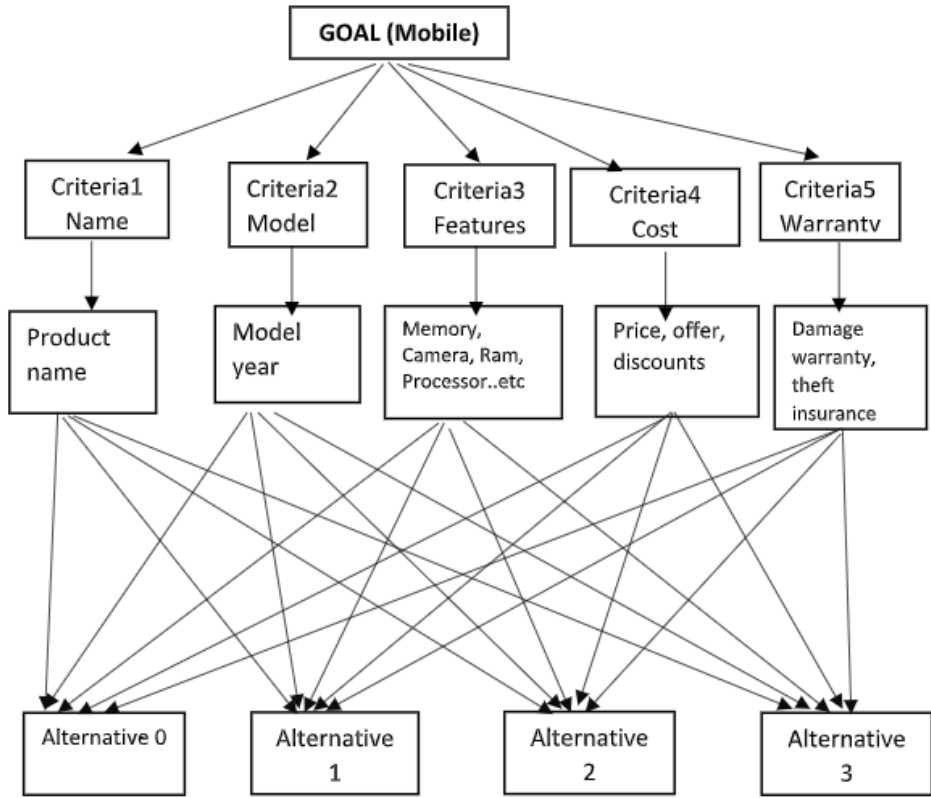

**Figure 2.** MCDM selction through AHP process for the alternatives.

**Table 3.** AHP Weight based on Expert Review.

| Comparisons | Importance in Scale |
|---|---|
| Name with model | 2 |
| Name with features | 5 |
| Name with cost | 4 |
| Name with warranty | 3 |
| Model with features | 4 |
| Model with cost | 3 |
| Model with warranty | 2 |
| Features with cost | 2 |
| Features with warranty | 3 |
| Cost with warranty | 2 |

Comparison of criteria is based on the importance we give. There are nine levels of importance in the AHP scale. Scale one indicates equal importance of both the criteria, three moderate importance, five strong importance, seven very strong importance, and nine extreme importance (two, four, six and eight are values in between).

5.1.2. Pairwise Comparisons

Pairwise comparison is a process of comparing entities in pairs to determine which entity has a greater amount of quantitative property. The pairwise comparison matrix table is the output which is based on the criteria priority we have provided. In the pairwise comparison matrix, the diagonals always get values of only one [33]. Based on the pairwise comparison table, a standardized matrix has been obtained. This table represents the weights of the criteria. The weight value is the average of attributes in the rows. Then, we represent the priorities in percentage. Based on the priority which has the highest percentage, we assign the rank. As per the analysis of the dataset which we have taken, the criteria which are named features get the highest priority, so we assign features that are

more important than the other criteria. We conclude that the criteria features are the first priority to select the best product based on the output of the AHP process.The pair-wise comparison matrix is presented in the Table 4. The ranking of the alternatives is presented in the Table 5. (The standardized matrix is determined from the pairwise comparison table by dividing the attributes with the sum of that column. For example, 1/15 = 0.06, 1 attribute of the first column; 15-sum of the first column. This is similar to the way we create the standardized matrix).

$$CI = \lambda - n/n - 1 \tag{2}$$

The consistency ration of the model CI is measured with the above Equation 2, where $n$ is the number of samples of the AHP matrix. The consistency matrix obtained for the prescribed work is 2.8. It should be less than 20 for the better and consistent importance matrix with correlation among the attributes.

**Table 4.** Pairwise Comparison Matrix.

| Name | Name | Model | Features | Cost | Warranty |
|------|------|-------|----------|------|----------|
| Name | 1 | 0.50 | 0.20 | 0.25 | 0.33 |
| Model | 2.00 | 1 | 0.25 | 0.33 | 0.50 |
| Features | 5.00 | 4.00 | 1 | 2.00 | 3.00 |
| Cost | 4.00 | 3.00 | 0.50 | 1 | 2.00 |
| Warranty | 3.00 | 2.00 | 0.33 | 0.50 | 1 |
| Sum | 15 | 10.5 | 2.28 | 4.08 | 6.83 |

**Table 5.** AHP Ranking of Alternatives.

| Name | Model | Features | Cost | Warranty | Priority | Rank |
|------|-------|----------|------|----------|----------|------|
| 0.06 | 0.04 | 0.08 | 0.06 | 0.04 | 5.60% | 5 |
| 0.13 | 0.09 | 0.1 | 0.08 | 0.07 | 9.40% | 4 |
| 0.33 | 0.38 | 0.43 | 0.49 | 0.43 | 41.20% | 1 |
| 0.26 | 0.28 | 0.21 | 0.24 | 0.29 | 25.60% | 2 |
| 0.2 | 0.19 | 0.14 | 0.12 | 0.14 | 15.80% | 3 |

*5.2. Estimation of the GRA Parameters*

5.2.1. Normalization of GRA

After collecting or generating the dataset, we have found maximum and minimum values for the taken criteria in the collected dataset. We have generated a normalization matrix for the collected dataset. The normalization matrix has two formulas: 1. the higher the better, 2. the lower the better. They are expressed in the Equations (3) and (4).

$$x = (xi(k) - \min(xi(k)))/(\max(xi(k)) - \min(xi(k))) \tag{3}$$

$$x = (\max(xi(k)) - xi(k)/(\max(xi(k)) - \min(xi(k))) \tag{4}$$

min $xi(k)$ is the current attribute, max $xi(k)$ is the maximum value of the criteria, min$xi(k)$ is the minimum value of the criteria. The maximum and minimum values of all the criteria have been calculated for each and every column. The normalized GRA matrix is presented in the Table 6. Normalization is the process of making the collected dataset into a normal set; it also looks like a simplified regular dataset [34]. In this process, we have used three criteria, i.e., model, features, warranty, and lower the better for one criteria, i.e., cost.

**Table 6.** GRA Normalized Matrix.

| Model | Features | Cost | Warranty |
|-------|----------|------|----------|
| 0 | 0.8 | 0.309523 | 0 |
| 0 | 0.65 | 0.777653 | 0 |
| 0.5 | 0.55 | 0.961773 | 0.5 |
| 1 | 0.65 | 0.20515 | 0 |
| 0.5 | 0.7 | 0.153321 | 0 |
| 1 | 0 | 0.512291 | 0 |
| 0 | 1 | 0.277512 | 1 |
| 1 | 0.45 | 0.949037 | 0.5 |
| 0 | 0.95 | 0.877044 | 0.5 |
| 0.5 | 0.55 | 0.025254 | 0 |
| 0 | 0.35 | 0.621494 | 0 |
| 1 | 0.45 | 0 | 1 |
| 0.5 | 0.75 | 0.372658 | 0 |
| 1 | 0.75 | 0.893959 | 0.5 |

### 5.2.2. Estimation of the Deviation Sequence

After calculating the normalization matrix, we have found maximum and minimum values for the criteria in the normalization matrix. Based on the normalization matrix, the deviation sequence matrix has been calculated. The maximum and minimum values in the deviation sequence matrix are known as delta maximum and delta minimum. We have also taken the zeta delta max value as 0.5, which is a constant value for all of the criteria.

Deviation Sequence Matrix is expressed as

$$x = xi(k) - xi'(k) \tag{5}$$

min *xi(k)* is the current attribute, max *xi(k)* is the maximum value of the criteria, which has been calculated for each and every column. The estimated values of the deviation sequence are tabulated in Table 7.

**Table 7.** Estimation of Deviation Sequence.

| Model | Features | Cost | Warranty |
|-------|----------|------|----------|
| 1 | 0.2 | 0.690477 | 1 |
| 0.5 | 0.3 | 0.500596 | 0.5 |
| 1 | 0.35 | 0.222347 | 1 |
| 0.5 | 0.45 | 0.038227 | 0.5 |
| 0 | 0.35 | 0.79485 | 1 |
| 0.5 | 0.3 | 0.846679 | 1 |
| 0 | 1 | 0.487709 | 1 |
| 1 | 0 | 0.722488 | 0 |
| 0 | 0.55 | 0.050963 | 0.5 |
| 1 | 0.05 | 0.122956 | 0.5 |
| 0.5 | 0.45 | 0.974746 | 1 |
| 1 | 0.65 | 0.378506 | 1 |
| 0 | 0.55 | 1 | 0 |
| 0.5 | 0.25 | 0.627342 | 1 |

### 5.2.3. Estimation of the Gray Relational Sequence

After calculating the deviation sequence matrix, we have found delta maximum and delta minimum values for the criteria in the deviation sequence matrix. We also have another value, which is called zeta delta max, in the deviation sequence matrix. Based on

the deviation sequence matrix, the grey relation coefficient table has been calculated as per the below Equation (6).

$$x = (\delta xi(k) + \xi \delta xi(k))/(xi(k) + \xi \delta x(k) \tag{6}$$

$-\delta$ min $xi(k)$ is the minimum value of the criteria, $\xi$ $\delta$ max $xi(k)$is the constant value which is 0.5, $xi(k)$ is the current attribute. Using the gray relation coefficient table, we have found the grade. The grade is the average value of all the criteria. Based on the grade values, the rank has been determined. The rank decides which is the best company product among all other products. The Grey Relational Grade is expressed in the Equation (7).

$$GRG = 1/n \sum [wk\xi_i(k)] \tag{7}$$

The above equation defines the Gray Relational Grades which provides the grades required for the ranking of the GRA process and the final results regarding the ranking of values are calculated using Equation (7). The final GRA ranks are represented in Table 8. The estimation of the deviation sequence is represented in Table 9. The appendix section contains the detailed description of GRA implementation in Tables A1–A4.

**Table 8.** Ranking through Gray Relational Co-Efficient.

| Model | Features | Cost | Warranty | Gray Grade | Rank |
|---|---|---|---|---|---|
| 1 | 0.714286 | 0.680617 | 1 | 0.848726 | 1 |
| 1 | 0.555556 | 0.775996 | 1 | 0.832888 | 2 |
| 1 | 0.47619 | 0.850498 | 1 | 0.831672 | 3 |
| 1 | 0.833333 | 0.413241 | 1 | 0.811644 | 4 |
| 1 | 0.47619 | 0.727893 | 1 | 0.801021 | 5 |
| 1 | 0.5 | 0.703387 | 1 | 0.800847 | 6 |
| 1 | 0.625 | 0.528423 | 1 | 0.788356 | 7 |
| 0.333333 | 0.769231 | 0.967763 | 1 | 0.767582 | 8 |
| 1 | 0.5 | 0.562163 | 1 | 0.765541 | 9 |
| 1 | 0.714286 | 0.343332 | 1 | 0.764405 | 10 |
| 1 | 0.909091 | 0.786545 | 0.333333 | 0.757242 | 11 |
| 1 | 0.5 | 0.521547 | 1 | 0.755387 | 12 |
| 1 | 0.666667 | 0.825027 | 0.5 | 0.747923 | 13 |
| 1 | 1 | 0.446196 | 0.5 | 0.736549 | 14 |
| 1 | 0.769231 | 0.822587 | 0.333333 | 0.731288 | 15 |

**Table 9.** Estimation of Deviation Sequence.

| Model | Features | Cost | Warranty |
|---|---|---|---|
| 1 | 0.2 | 0.690477 | 1 |
| 0.5 | 0.3 | 0.500596 | 0.5 |
| 1 | 0.35 | 0.222347 | 1 |
| 0.5 | 0.45 | 0.038227 | 0.5 |
| 0 | 0.35 | 0.79485 | 1 |
| 0.5 | 0.3 | 0.846679 | 1 |
| 0 | 1 | 0.487709 | 1 |
| 1 | 0 | 0.722488 | 0 |
| 0 | 0.55 | 0.050963 | 0.5 |
| 1 | 0.05 | 0.122956 | 0.5 |
| 0.5 | 0.45 | 0.974746 | 1 |
| 1 | 0.65 | 0.378506 | 1 |
| 0 | 0.55 | 1 | 0 |
| 0.5 | 0.25 | 0.627342 | 1 |
| 0 | 0.25 | 0.106041 | 0.5 |

## 6. Discussion

This section describes security mechanisms that can be applied to the dataset during the pre-recommendation and post-recommendation process. This ensures the encryption of the desired attribute on the dataset to provide security to withstand unauthorized access by competitors and others. This section describes the access-based security model followed by the implementation of the Environment-Based Attribute Access Control model algorithm.

### 6.1. Access-Based Security Control Mechanism

Safeguarding assets becomes a top priority for a system such as the one we have proposed. Access might be limited to certain users as a remedy for this. Given that it is reliable, a system that can identify authorized users and bar access to illicit items can be effective [35]. To aid this, we propose a comprehensive access control strategy that supports read and written access for record actors, as well as enforcing framework-level encryption-related activities. To further develop the role-based access control model, we support role hierarchies that allow roles to be organized hierarchically. Less privileged positions are at lower levels of the hierarchy and highest roles are at the top, as described in the algorithm below. Our approach maps specific roles in the EBAACM model to specific sets of attributes adding on to it, and higher roles could inherit all permissions from lower roles. For example, someone who contributes significantly to configuring and approving phone features has a higher role than someone who only pulls information from the system. Additionally, the owner of the system who maintains the dataset has higher priority than anyone else simply exploiting certain feature comparison through the model. Based on our system, the model has few process steps which help achieve robustness, which are described as follows:

1. Environment specification—Static or Dynamic state space (completely based on role hierarchy)
2. For each user who logs in and utilizes the system for their own needs, a digital ID should be created. A user or someone who wants to be part of a fuzzy ecosystem generates a random number $s_k$ as a private key, generates the corresponding $p_k$ as a public key and chooses a secure hash function followed by double hashing the public key.

$$H : \{0,1\}\star \to \{0,1\}\lambda \tag{8}$$

   This is a unique identifier for each user to track their operations and transactions.

3. The proposed model defines attributes for the role-based access control model [21]. The model also defines who owns the data and who is the user of the data. Each entity requesting access to a record can be evaluated against a specially designed hierarchy according to specific criteria or its position.
4. Environment-Based Attribute Access Control model (EBAACM). This algorithm designed to protect the environment, as discussed further.
5. External attributes are defined, such as time of operation, location, working hours with the dataset, etc.
6. The Certificate Authority (CA) is a trusted authority that issues public key certificates (X.509 certificates) [36] to all entities, including users, AAs, data owners and proxies. Key pairs and certificates are used to sign transactions, encrypt private keys and authenticate entities within the system.
7. Proxy is a semi-trusted server responsible for decrypting and re-encrypting data [37]. In our design, the proxy server has its own PKI key pair and installed X.509 certificates that are used to authenticate other system modules.

### 6.2. Environment-Based Attribute Access Control Model (EBAACM) Algorithm

The dataset must be protected and authenticated; hence, rigorous data access control permissions must be set. A secure environment-based attribute access-control paradigm is suggested for the same reason. The environment must first be initialized in order to set

the security services of the model we constructed. The framework ecosystem has been split into two categories: static and dynamic. Users with the lowest role, such as those looking for recommendations, access information in a static environment. This audience will only be permitted to obtain legal information; no other transactions will be permitted. In a dynamic state, different operations are carried out by employees according to the importance accorded to each position. They have a variety of tasks to complete, such as updating and adding to datasets. The static and dynamic environments are connected to certain environmental characteristics that relate to access times, working and non-working hours, locations, or dynamic elements of the access control scenario. The architecture of the system is represented in Figure 3. These are linked to the security-related rules created for the corporation. By initially stating the kind of account the user owns, all of the above is implemented, along with a more reliable security procedure. The development of a digital identity is the first step in this process. Adding to this, some characteristics help to determine a person's intent and the environment's integrity. The key used to guarantee safe data transmission between the server and the client is $s_k$. The user shares its digital account identity and the symmetric key$s_k$ and the corresponding data information can be obtained by decrypting the key $s_k$. Various functions used in the algorithm, such as hasIssueRole, renounceIssueRole and parExtension, help the framework in achieving a secured space to function. After the digital identity is authenticated and a role is identified for it, the model can access the framework accordingly. Each of the entity's transactions is taken into account, along with its authorization. Therefore, a secured environment for the fuzzy framework is achieved. The environment based access control algorithm is represented in Table 10.

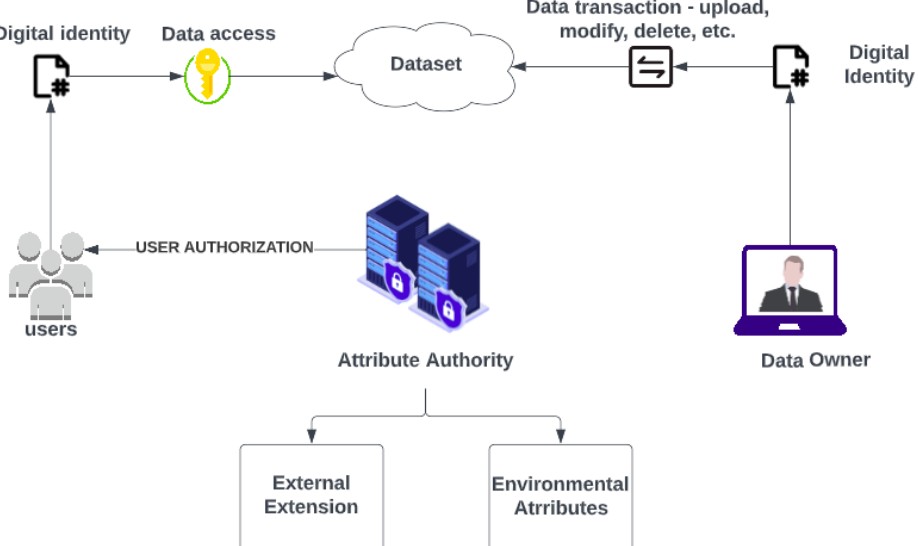

**Figure 3.** Architecture of the Environment-Based Attribute Access Control model(EBAACM) Algorithm.

**Table 10.** Environment-Based Attribute Access Control model(EBAACM) Algorithm.

| Step | Steps of Algorithm |
|---|---|
| 1 | Establish user authorization. |
| 2 | function hasIssueRole(role,account) public view |
| 3 | require(account! = address(0), "Roles: this account is the zero address"); |
| 4 | return role.bearer[account]; |
| 5 | end function |
| 6 | Assign authorization to user account |
| 7 | nction addIssueRole(role, account) public onlyOwner |
| 8 | require(!has(role, account), "Roles: this account already has role"); |
| 9 | role.bearer[account] = false; |
| 10 | end function |
| 11 | Dataset authority revokes role authorization |
| 12 | function renounceIssueRole(role,account) public onlyOwner |
| 13 | require(has(role, account), "Roles: this account does not have role"); |
| 14 | role.bearer[account] = false; |
| 15 | Extending the access structure externally |
| 16 | function par Extension(role,account) |
| 17 | require(has(system parameters,role,extended role), "Roles: this account role has been extended"); |
| 18 | role.bearer[account] = false; |
| 19 | end function |

*6.3. Sensitivity Analysis*

The GRA results are not weight normalized but min and max normalized. Hence, the sensitivity analysis is first carried out on the AHP implementation, where the weight change can be followed up with a change in the Consistency Ratio (CR) and the final ranking. However, as per the original assessment of the AHP method based on the expert opinion, the features were ranked as per the importance: Features > Cost > Warranty > Model.

This assumption produced a consistency ratio of 1.5% for the attributes under subjective weight assumptions.The original feature importance distribution is shown in Figure 4.

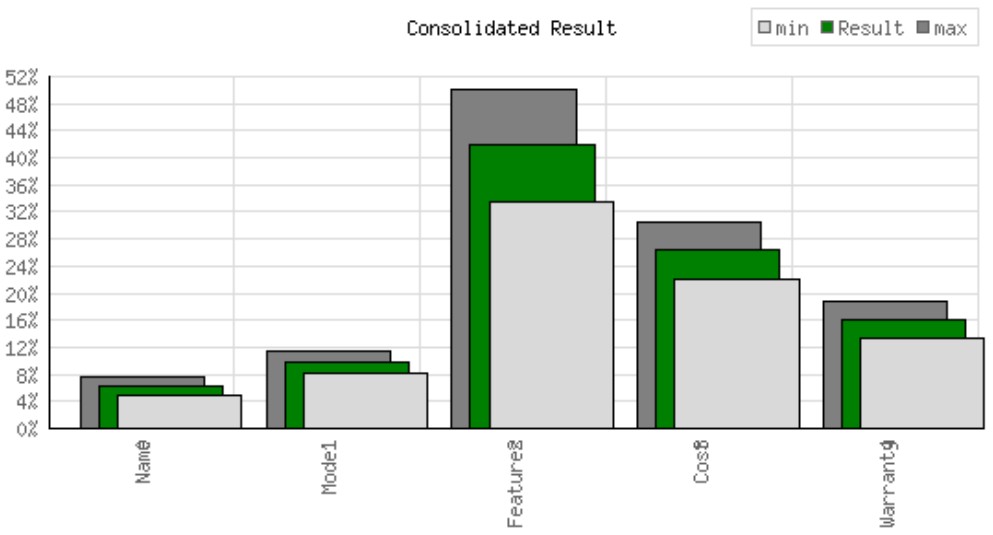

**Figure 4.** Attributes with Weight Representation in AHP.

The sensitivity analysis for GRA is discussed with the Figure 5. The diagram shows how the attributes contribute to the estimation of the Gray Relational Grade and the final ranking. The rank is expressed in the X-axis and the corresponding values of the grade

are represented in the Y-axis in terms of percentage. As per the contribution, the feature importance follows the following order: Warranty > Cost > features > Model.

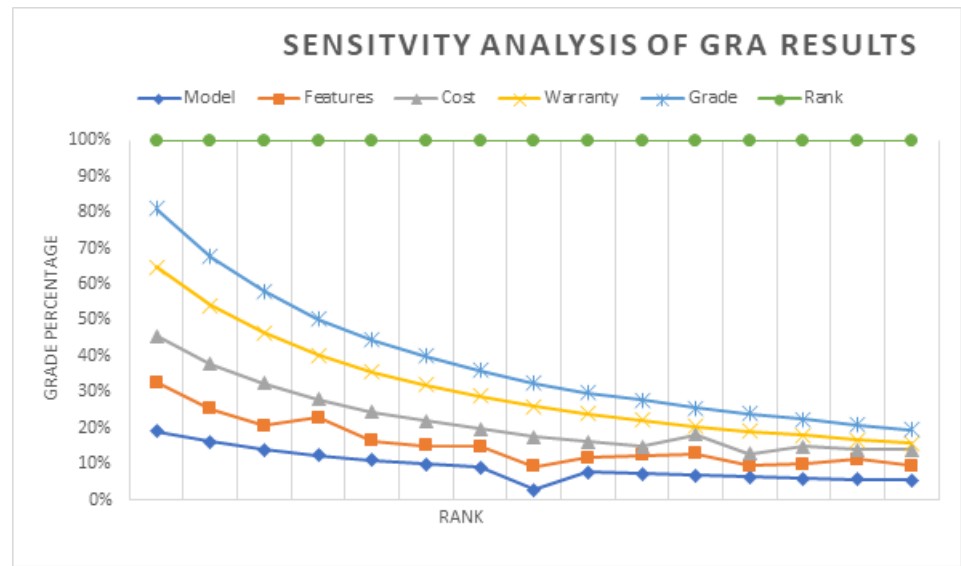

**Figure 5.** Attributes with Weight Representation in AHP.

The Warranty is the primary contributing factor for the highest rank, comprising 70%, followed by cost at around 50%, then features and model around 35% and 20%, respectively.

### 6.4. Validation

The GRA results are not subjective but objective, and are estimated from the Gray Relational co-efficient after normalization through min–max analysis. However, as per the original assessment of AHP method based on an expert opinion, the features were ranked as per the importance, as follows: Features > Cost > Warranty > Model. This assumption produced a consistency ratio of 1.5% for the attributes under subjective weight assumptions. However, post-GRA process, the AHP weights were re-assigned based on the outcomes obtained from the GRA ranking and sensitivity analysis. The change in importance is presented in the Table 11. The order of weights and the final ranking is represented along with the features in Figure 6.

**Table 11.** AHP Weight based on Expert Review.

| Comparisons | Importance in Scale |
| --- | --- |
| Model with Name | 2 |
| Features with Name | 3 |
| Cost With Name | 4 |
| Warranty With Name | 5 |
| Features with model | 2 |
| Cost with model | 3 |
| Warranty with model | 4 |
| Cost with features | 2 |
| Warrant with features | 3 |
| Warranty with cost | 2 |

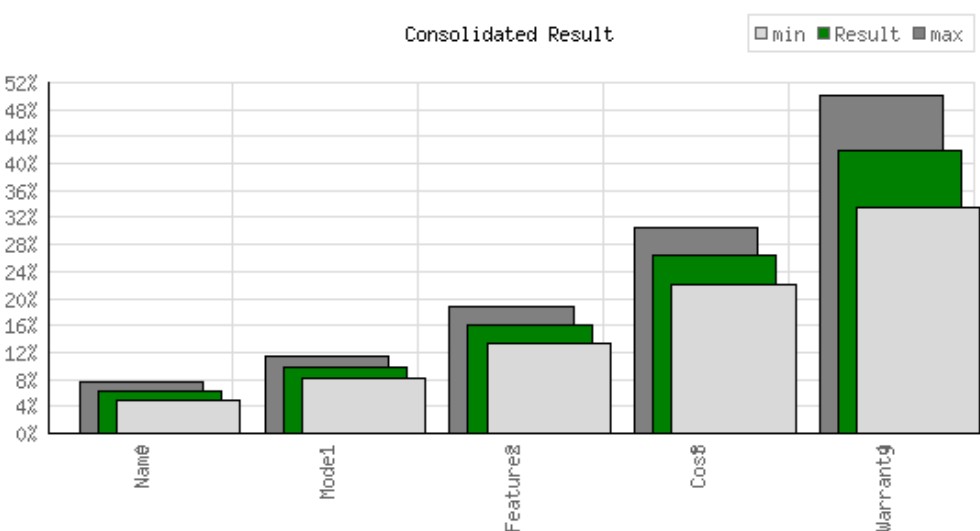

**Figure 6.** Attributes with Weight Representation in AHP.

The revised weight distribution and the importance are shown in the Figure 6. From this observation, we understand that the consistency ratio remains as 1.5%, which is similar to the one which we received at the initial implementation of AHP through the inputs received from the subject-matter experts. This validates the results even after the change of weights produced a similar consistency ratio and does not impact the prediction made by GRA. The output weights match the subject weights; thus, the results are balanced with the original subjective assumption.

*6.5. Limitations of the Proposed System*

The limitations of the proposed work are listed below,

- The AHP results are subjective, but GRA results are not subjective.
- There may be a need to re-assign the weights of AHP after the estimation of GRA.
- In such a case, there may be a change in the Consistency Ratio based on the weight change.
- So, we need to deploy different MCDM techniques if the consistency ratio changes.
- This work deals only with attribute-based encryption, which may be extended to data or transaction-based encryption.

**7. Conclusions and Further Work**

There are many research papers on the basis of AHP and GRA, which represent the production process of the products. However, in this paper we have found the best process of all. We have used both AHP and GRA processes along with security mechanism implemented for the dataset. As per the AHP process, we have concluded that the features of the product are more important than the other criteria. So, on the basis of output of the AHP process, we conclude that the features are important to select the product. The whole framework is secured using the EBAACM algorithm that utilizes an access control methodology and various other essential environmental factors to set access permissions for all digitally identified users.

*Future Directions*

- The business intelligence enhancements can be deployed across various MCDM models such as VIKOR, WASPAS, MOORA.
- The sensitivity analysis of various MCDM models can be compared to improve our understanding of the real competitive advantage of business intelligence.
- Attribute-based security mechanisms can be enhanced with the transaction-based encryption with policies.
- Deep learning models can be used for the decision-support systems with real-time business-related big data analysis.

The intricacy of context data will be a topic of future investigation. In this direction, crowdsourcing can be used to gather structured and unstructured data from many sources with more individualized qualities and metadata. Due to the computational requirements of even a moderately sized problem, the system presented in this investigation places significant constraints on the scope of the research. One limitation of AHP is that the modeling process itself is inherently subjective. That means the approach cannot say for sure that the verdicts are correct. Moreover, once the datasets gathered via crowdsourcing technique grow more complicated, numerous unsolved security issues may hinder the performance of the system. This will make it possible to better understand users and deliver more individualized outcomes. Utilizing vast quantities of context data, more effort is required in terms of safeguarding user privacy in addition to data security, while still maintaining adequate recommender systems speed and accuracy.

**Author Contributions:** Conceptualization, M.K.N. and I.J.; methodology, M.K.N., I.J.; software, M.K.N.; validation: S.S.; formal analysis, M.K.N.; investigation S.B. and A.M.A.; resources, M.K.N. and I.J.; data curation, A.A.M.; writing—original draft preparation, M.K.N., I.J., S.S.; writing—review and editing; S.S., A.M.A. and S.B.; visualization, I.J.; supervision, S.S.; project administration, S.B.; funding acquisition, S.B. and A.A.M. All authors have read and agreed to the published version of the manuscript.

**Funding:** This research received no external funding.

**Institutional Review Board Statement:** Not Applicable.

**Informed Consent Statement:** Not Applicable.

**Data Availability Statement:** Not Applicable.

**Acknowledgments:** Princess Nourah bint Abdulrahman University Researchers Supporting Project number (PNURSP2023R151), Princess Nourah bint Abdulrahman University, Riyadh, Saudi Arabia.

**Conflicts of Interest:** The authors declare no conflicts of interest.

## Appendix A. Experimental Analysis of the GRA

*GRA Implementation Steps*

This section contains detailed implementation tables for GRA. The sample of the same had already been provided in the main document. This section contains tables related to GRA Normalization, Estimation of the deviation sequence, Estimation of the Grey Relational co-efficient and finally the grey relational grades and ranks.

**Table A1.** GRA Normalized Matrix.

| Model | Features | Cost | Warranty |
|---|---|---|---|
| 0 | 0.8 | 0.309523 | 0 |
| 0 | 0.65 | 0.777653 | 0 |
| 0.5 | 0.55 | 0.961773 | 0.5 |
| 1 | 0.65 | 0.20515 | 0 |
| 0.5 | 0.7 | 0.153321 | 0 |
| 1 | 0 | 0.512291 | 0 |
| 0 | 1 | 0.277512 | 1 |
| 1 | 0.45 | 0.949037 | 0.5 |
| 0 | 0.95 | 0.877044 | 0.5 |
| 0.5 | 0.55 | 0.025254 | 0 |
| 0 | 0.35 | 0.621494 | 0 |
| 1 | 0.45 | 0 | 1 |
| 0.5 | 0.75 | 0.372658 | 0 |
| 1 | 0.75 | 0.893959 | 0.5 |
| 0 | 1 | 0.364601 | 1 |
| 0.5 | 0.1 | 0.224665 | 0.5 |
| 0.5 | 0.35 | 0.215828 | 0 |
| 1 | 0 | 0.845098 | 0 |
| 0.5 | 0.45 | 0.325356 | 0 |
| 0 | 0.5 | 0.147257 | 1 |
| 1 | 0.05 | 0.80204 | 0.5 |
| 0 | 1 | 0.879773 | 0 |
| 1 | 0.45 | 0.912109 | 1 |
| 0 | 0.2 | 0.607264 | 0 |
| 0.5 | 0 | 0.304282 | 0 |
| 1 | 0.4 | 1 | 0 |
| 0 | 0.15 | 0.221676 | 0.5 |
| 0.5 | 0.25 | 0.927639 | 1 |
| 0.5 | 0.8 | 0.588898 | 0.5 |
| 0 | 0.4 | 0.5064 | 0.5 |
| 1 | 0.6 | 0.349678 | 0 |
| 0 | 0.3 | 0.572134 | 0 |
| 1 | 0.45 | 0.315804 | 1 |
| 1 | 0.5 | 0.16937 | 0.5 |
| 0 | 0.75 | 0.500162 | 0.5 |
| 0 | 0.7 | 0.496394 | 0.5 |
| 0.5 | 0.2 | 0.378203 | 0.5 |
| 0 | 0.9 | 0.660393 | 1 |
| 1 | 1 | 0.379416 | 0.5 |
| 1 | 0.8 | 0.043685 | 1 |
| 0 | 0.1 | 0.794178 | 0 |
| 1 | 0.95 | 0.864309 | 0 |
| 1 | 0.45 | 0.813086 | 1 |
| 1 | 0.7 | 0.553789 | 1 |
| 1 | 0.6 | 0.855667 | 1 |
| 1 | 0.55 | 0.266531 | 0 |
| 1 | 0.8 | 0.765372 | 1 |

**Table A2.** EStimation of Deviation Sequence.

| Model | Features | Cost | Warranty |
|---|---|---|---|
| 1 | 0.2 | 0.690477 | 1 |
| 0.5 | 0.3 | 0.500596 | 0.5 |
| 1 | 0.35 | 0.222347 | 1 |
| 0.5 | 0.45 | 0.038227 | 0.5 |
| 0 | 0.35 | 0.79485 | 1 |
| 0.5 | 0.3 | 0.846679 | 1 |
| 0 | 1 | 0.487709 | 1 |
| 1 | 0 | 0.722488 | 0 |
| 0 | 0.55 | 0.050963 | 0.5 |
| 1 | 0.05 | 0.122956 | 0.5 |
| 0.5 | 0.45 | 0.974746 | 1 |
| 1 | 0.65 | 0.378506 | 1 |
| 0 | 0.55 | 1 | 0 |
| 0.5 | 0.25 | 0.627342 | 1 |
| 0 | 0.25 | 0.106041 | 0.5 |
| 1 | 0 | 0.635399 | 0 |
| 0.5 | 0.9 | 0.775335 | 0.5 |
| 0.5 | 0.65 | 0.784172 | 1 |
| 0 | 1 | 0.154902 | 1 |
| 0.5 | 0.55 | 0.674644 | 1 |
| 1 | 0.5 | 0.852743 | 0 |
| 0 | 0.95 | 0.19796 | 0.5 |
| 1 | 0 | 0.120227 | 1 |
| 0 | 0.55 | 0.087891 | 0 |
| 1 | 0.8 | 0.392736 | 1 |
| 0.5 | 1 | 0.695718 | 1 |
| 0 | 0.6 | 0 | 1 |
| 1 | 0.85 | 0.778324 | 0.5 |
| 0.5 | 0.75 | 0.072361 | 0 |
| 0.5 | 0.2 | 0.411102 | 0.5 |
| 1 | 0.6 | 0.4936 | 0.5 |
| 0 | 0.4 | 0.650322 | 1 |
| 1 | 0.7 | 0.427866 | 1 |
| 0 | 0.55 | 0.684196 | 0 |
| 0 | 0.5 | 0.83063 | 0.5 |
| 1 | 0.25 | 0.499838 | 0.5 |
| 1 | 0.3 | 0.503606 | 0.5 |
| 0.5 | 0.8 | 0.621797 | 0.5 |
| 1 | 0.1 | 0.339607 | 0 |
| 0 | 0 | 0.620584 | 0.5 |
| 0 | 0.2 | 0.956315 | 0 |
| 1 | 0.9 | 0.205822 | 1 |
| 0 | 0.05 | 0.135691 | 1 |
| 0 | 0.55 | 0.186914 | 0 |
| 0 | 0.3 | 0.446211 | 0 |
| 0 | 0.4 | 0.144333 | 0 |
| 0 | 0.45 | 0.733469 | 1 |

**Table A3.** Ranking through Grey Relational Co-Efficient.

| Model | Features | Cost | Warranty | Grey Grade | Rank |
|-------|----------|----------|----------|------------|------|
| 1 | 0.714286 | 0.680617 | 1 | 0.848726 | 1 |
| 1 | 0.555556 | 0.775996 | 1 | 0.832888 | 2 |
| 1 | 0.47619 | 0.850498 | 1 | 0.831672 | 3 |
| 1 | 0.833333 | 0.413241 | 1 | 0.811644 | 4 |
| 1 | 0.47619 | 0.727893 | 1 | 0.801021 | 5 |
| 1 | 0.5 | 0.703387 | 1 | 0.800847 | 6 |
| 1 | 0.625 | 0.528423 | 1 | 0.788356 | 7 |
| 0.333333 | 0.769231 | 0.967763 | 1 | 0.767582 | 8 |
| 1 | 0.5 | 0.562163 | 1 | 0.765541 | 9 |
| 1 | 0.714286 | 0.343332 | 1 | 0.764405 | 10 |
| 1 | 0.909091 | 0.786545 | 0.333333 | 0.757242 | 11 |
| 1 | 0.5 | 0.521547 | 1 | 0.755387 | 12 |
| 1 | 0.666667 | 0.825027 | 0.5 | 0.747923 | 13 |
| 1 | 1 | 0.446196 | 0.5 | 0.736549 | 14 |
| 1 | 0.769231 | 0.822587 | 0.333333 | 0.731288 | 15 |
| 1 | 0.47619 | 0.422228 | 1 | 0.724604 | 16 |
| 1 | 0.47619 | 0.907502 | 0.5 | 0.720923 | 17 |
| 0.333333 | 1 | 0.539009 | 1 | 0.718086 | 18 |
| 1 | 0.384615 | 0.473466 | 1 | 0.71452 | 19 |
| 1 | 0.47619 | 0.333333 | 1 | 0.702381 | 20 |
| 1 | 0.46546 | 0.333333 | 0.8 | 0.69997 | 21 |
| 1 | 0.454545 | 1 | 0.333333 | 0.69697 | 22 |
| 0.5 | 0.47619 | 0.801316 | 1 | 0.694377 | 23 |
| 0.333333 | 1 | 0.440374 | 1 | 0.693427 | 24 |
| 0.5 | 0.4 | 0.873574 | 1 | 0.693393 | 25 |
| 0.333333 | 0.833333 | 0.595517 | 1 | 0.690546 | 26 |
| 0.333333 | 1 | 0.409002 | 1 | 0.685584 | 27 |
| 0.333333 | 0.454545 | 0.935071 | 1 | 0.680737 | 28 |
| 0.5 | 0.588235 | 0.625683 | 1 | 0.678479 | 29 |
| 0.5 | 0.714286 | 0.993288 | 0.5 | 0.676893 | 30 |
| 1 | 0.714286 | 0.488504 | 0.5 | 0.675698 | 31 |
| 1 | 0.555556 | 0.807482 | 0.333333 | 0.674093 | 32 |
| 0.5 | 0.625 | 0.570555 | 1 | 0.673889 | 33 |
| 0.333333 | 0.666667 | 0.688636 | 1 | 0.672159 | 34 |
| 1 | 0.434783 | 0.875927 | 0.333333 | 0.661011 | 35 |
| 1 | 0.344828 | 0.744946 | 0.5 | 0.647443 | 36 |
| 1 | 0.714286 | 0.374762 | 0.5 | 0.647262 | 37 |
| 0.5 | 0.416667 | 0.669387 | 1 | 0.646514 | 38 |
| 0.5 | 0.625 | 0.956932 | 0.5 | 0.645483 | 39 |
| 1 | 0.344828 | 0.716374 | 0.5 | 0.6403 | 40 |
| 0.333333 | 0.909091 | 0.802625 | 0.5 | 0.636262 | 41 |
| 0.5 | 0.454545 | 0.556686 | 1 | 0.627808 | 42 |
| 1 | 0.5 | 0.480622 | 0.5 | 0.620156 | 43 |
| 0.333333 | 1 | 0.806156 | 0.333333 | 0.618206 | 44 |
| 1 | 0.5 | 0.628339 | 0.333333 | 0.615418 | 45 |
| 0.5 | 0.526316 | 0.928975 | 0.5 | 0.613823 | 46 |
| 0.5 | 0.384615 | 0.552331 | 1 | 0.609237 | 47 |

**Table A4.** Estimation of Deviation Sequence.

| Model | Features | Cost | Warranty |
|:---:|:---:|:---:|:---:|
| 1 | 0.2 | 0.690477 | 1 |
| 0.5 | 0.3 | 0.500596 | 0.5 |
| 1 | 0.35 | 0.222347 | 1 |
| 0.5 | 0.45 | 0.038227 | 0.5 |
| 0 | 0.35 | 0.79485 | 1 |
| 0.5 | 0.3 | 0.846679 | 1 |
| 0 | 1 | 0.487709 | 1 |
| 1 | 0 | 0.722488 | 0 |
| 0 | 0.55 | 0.050963 | 0.5 |
| 1 | 0.05 | 0.122956 | 0.5 |
| 0.5 | 0.45 | 0.974746 | 1 |
| 1 | 0.65 | 0.378506 | 1 |
| 0 | 0.55 | 1 | 0 |
| 0.5 | 0.25 | 0.627342 | 1 |
| 0 | 0.25 | 0.106041 | 0.5 |
| 1 | 0 | 0.635399 | 0 |
| 0.5 | 0.9 | 0.775335 | 0.5 |
| 0.5 | 0.65 | 0.784172 | 1 |
| 0 | 1 | 0.154902 | 1 |
| 0.5 | 0.55 | 0.674644 | 1 |
| 1 | 0.5 | 0.852743 | 0 |
| 0 | 0.95 | 0.19796 | 0.5 |
| 1 | 0 | 0.120227 | 1 |
| 0 | 0.55 | 0.087891 | 0 |
| 1 | 0.8 | 0.392736 | 1 |
| 0.5 | 1 | 0.695718 | 1 |
| 0 | 0.6 | 0 | 1 |
| 1 | 0.85 | 0.778324 | 0.5 |
| 0.5 | 0.75 | 0.072361 | 0 |
| 0.5 | 0.2 | 0.411102 | 0.5 |
| 1 | 0.6 | 0.4936 | 0.5 |
| 0 | 0.4 | 0.650322 | 1 |
| 1 | 0.7 | 0.427866 | 1 |
| 0 | 0.55 | 0.684196 | 0 |
| 0 | 0.5 | 0.83063 | 0.5 |
| 1 | 0.25 | 0.499838 | 0.5 |
| 1 | 0.3 | 0.503606 | 0.5 |
| 0.5 | 0.8 | 0.621797 | 0.5 |
| 1 | 0.1 | 0.339607 | 0 |
| 0 | 0 | 0.620584 | 0.5 |
| 0 | 0.2 | 0.956315 | 0 |
| 1 | 0.9 | 0.205822 | 1 |
| 0 | 0.05 | 0.135691 | 1 |
| 0 | 0.55 | 0.186914 | 0 |
| 0 | 0.3 | 0.446211 | 0 |
| 0 | 0.4 | 0.144333 | 0 |
| 0 | 0.45 | 0.733469 | 1 |

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
