# Peer review of "Secured MCDM Model for Crowdsource Business Intelligence"

_applsci, doi:10.3390/app13031511_

Round 1
Reviewer 1 Report
Greetings,
The authors have done a good job on the paper. It is necessary to make certain corrections. In the introduction, you wrote what the contribution of the research is, you should emphasize what gaps this research solves and what question this research answers. The contribution is good compared to other similar researches. The literature review is good and the references are good and all up to date. Before explaining the selection segments 3. Recent Research on Crowd-sourcing with AI, it is necessary to first explain those segments and then explain them individually. After the selection of the 4th Architecture of the System, it is necessary to explain the AHP method used in this research. The explanation should not be in the results but before. So before doing the results, it is necessary to explain the methods first. In the selection, the results should only be the results and their explanation, not the explanation of the methods. Other parts of the paper are well done. The boards are good. The references used are good. It is only necessary to emphasize the limitation of this research.
All the best.
Author Response
Reviewer 1 Comments and Responses
Comments: The authors have done a good job on the paper. It is necessary to make certain corrections. In the introduction, you wrote what the contribution of the research is, you should emphasize what gaps this research solves and what question this research answers. The contribution is good compared to other similar researches. The literature review is good and the references are good and all up to date. Before explaining the selection segments 3. Recent Research on Crowd-sourcing with AI, it is necessary to first explain those segments and then explain them individually. After the selection of the 4th Architecture of the System, it is necessary to explain the AHP method used in this research. The explanation should not be in the results but before. So before doing the results, it is necessary to explain the methods first. In the selection, the results should only be the results and their explanation, not the explanation of the methods. Other parts of the paper are well done. The boards are good. The references use
Reviewer 1 Comments and Responses
Comments: The authors have done a good job on the paper. It is necessary to make certain corrections. In the introduction, you wrote what the contribution of the research is, you should emphasize what gaps this research solves and what question this research answers. The contribution is good compared to other similar researches. The literature review is good and the references are good and all up to date. Before explaining the selection segments 3. Recent Research on Crowd-sourcing with AI, it is necessary to first explain those segments and then explain them individually. After the selection of the 4th Architecture of the System, it is necessary to explain the AHP method used in this research. The explanation should not be in the results but before. So before doing the results, it is necessary to explain the methods first. In the selection, the results should only be the results and their explanation, not the explanation of the methods. Other parts of the paper are well done. The boards are good. The references used are good. It is only necessary to emphasize the limitation of this research.
Response: Thank you for your suggestions. The introduction section has been modified as per your suggestions focusing on research gaps and solutions. Recent Research on Crowd-sourcing with AI has been explained. AHP method has been explained in the chapter of architecture of the system. The other methods are explained before the results.
d are good. It is only necessary to emphasize the limitation of this research.
Response: Thank you for your suggestions. The introduction section has been modified as per your suggestions focusing on research gaps and solutions. Recent Research on Crowd-sourcing with AI has been explained. AHP method has been explained in the chapter of architecture of the system. The other methods are explained before the results.
Reviewer 2 Report
Authors have presented “”Secured MCDM model for Crowdscource Business Intelligence The idea is very interested and well presented”.
I think the manuscript needs a revision before going to the next step. The following are the suggestions to include in the manuscript.
1) The abstract should be written by focusing only on the objective of the work.
2) In introduction, authors should add the information related to AHP and GRA both processes and cite following important articles related to decision making to support the readability and practicality.
(a) decision-making technique and its application for assessment problem.
(b) A novel approach to generalized to AHP and GRA both processes and its application in Crowdscource Business Intelligence
3) The contributions of the paper in Introduction Section should be clearly mentioned.
4) Conclusion section should be rewritten by adding the advantages and contribution of the work, Add the future research work direction in the separate paragraph.
5) There was no sensitivity analysis done. Prior to the conclusion section, this procedure should be completed.
6) Reference section needs to be polish by adding some recent articles related to diverse approaches. Several references have incomplete details. Authors should update it completely and properly.
https://doi.org/10.3390/su14031118
https://doi.org/10.3390/su14084835
7) Future research direction should be elaborated in more detail.
Author Response
Reviewer 2 Comments and Responses
Comment 1: The abstract should be written by focusing only on the objective of the work.
Responses: Thanks for the suggestions. The abstract has been modified accordingly.
Comment 2: In introduction, authors should add the information related to AHP and GRA both processes and cite following important articles related to decision making to support the readability and practicality.
(a) decision-making technique and its application for assessment problem.
(b) A novel approach to generalized to AHP and GRA both processes and its application in Crowdscource Business Intelligence
Responses: Thanks for the suggestions. The introduction has been updated with the suggestions.
Comments 3: The contributions of the paper in Introduction Section should be clearly mentioned.
Responses: Thanks for the suggestions. A subsection addressing contributions of work has been added below the introduction.
Comment 4: Conclusion section should be rewritten by adding the advantages and contribution of the work, Add the future research work direction in the separate paragraph.
Responses: Thanks for the suggestions. Conclusion has been rewritten and future work has been written separately.
Comments 5: There was no sensitivity analysis done. Prior to the conclusion section, this procedure should be completed.
Responses: Thanks for the suggestions. Sensitivity analysis is done as a separate subsection under discussion.
Comment 6: Reference section needs to be polish by adding some recent articles related to diverse approaches. Several references have incomplete details. Authors should update it completely and properly.
Responses: Thanks for the suggestions. Recent references has been added and references are updated.
Reviewer 3 Report
1. The contributions are not meaningful. This subsection must be rewritten
2. Literature review must be presented in a table format to improve the clarity of the presentation.
3. Research questions must be elaborated on.
4. The motivation for selecting AHP needs to be explained.
5. Why GRA? Its pros and cons must be discussed.
6. Many long tables should be moved to the Appendix.
7. The authors did not validate their results. Why?
8. Sensitivity analyses must be added.
9. This topic is very interesting for authorities. Practical implications need to be added as a separate sub-section.
10. Limitations of the research should be provided in the last section.
11. The language is poor. Professional editing is strongly suggested.
Author Response
Reviewer 3 comments and responses
Comment 1: The contributions are not meaningful. This subsection must be rewritten
Responses: Thanks for the suggestions. The contributions has been updated in a separate subsection
Comment 2:. Literature review must be presented in a table format to improve the clarity of the presentation.
Responses: Thanks for the suggestions. A summary table has been added.
Comment 3: Research questions must be elaborated on.
Responses: Thanks for the suggestions. Research gaps has been explained under subsection contribution of the work
Comment 4: The motivation for selecting AHP needs to be explained.
Responses: Thanks for the suggestions. Roles of AHP and GRA and advantages have been explained under subsections 1.3 and 1.4.
Comment 5: Why GRA? Its pros and cons must be discussed.
Responses: Thanks for the suggestions. Roles of AHP and GRA and advantages have been explained under subsections 1.3 and 1.4.
Comment 6: Many long tables should be moved to the Appendix.
Responses: Thanks for the suggestions. Long table has been moved to Appendix.
Comment 7: The authors did not validate their results. Why?
Responses: Thanks for the suggestions. Sensitivity analysis and validation has been done under the results
Comment 8: Sensitivity analyses must be added.
Responses: Thanks for the suggestions. Sensitivity analysis and validation has been done under the results
Comment 9: This topic is very interesting for authorities. Practical implications need to be added as a separate sub-section.
Responses: Thanks for the suggestions. Sensitivity analysis and validation has been done under the results
Comment 10: Limitations of the research should be provided in the last section.
Responses: Thanks for the suggestions. Limitations has been added under results section.
Comment 11: The language is poor. Professional editing is strongly suggested.
Responses: Thanks for the suggestions. Thorough proof read has been done.
Round 2
Reviewer 2 Report
Accept as it is
Reviewer 3 Report
The authors significantly improved the paper. I do not have any further comments.